# Giant Kerr response of ultrathin gold films from quantum size effect

Haoliang Qian[1,*], Yuzhe Xiao[1,*] & Zhaowei Liu[1,2,3]

With the size of plasmonic devices entering into the nanoscale region, the impact of quantum physics needs to be considered. In the past, the quantum size effect on linear material properties has been studied extensively. However, the nonlinear aspects have not been explored much so far. On the other hand, much effort has been put into the field of integrated nonlinear optics and a medium with large nonlinearity is desirable. Here we study the optical nonlinear properties of a nanometre scale gold quantum well by using the z-scan method and nonlinear spectrum broadening technique. The quantum size effect results in a giant optical Kerr susceptibility, which is four orders of magnitude higher than the intrinsic value of bulk gold and several orders larger than traditional nonlinear media. Such high nonlinearity enables efficient nonlinear interaction within a microscopic footprint, making quantum metallic films a promising candidate for integrated nonlinear optical applications.

[1] Department of Electrical and Computer Engineering, University of California, San Diego, 9500 Gilman Drive, La Jolla, California 92093, USA. [2] Materials Science and Engineering, University of California, San Diego, 9500 Gilman Drive, La Jolla, California 92093, USA. [3] Center for Memory and Recording Research, University of California, San Diego, 9500 Gilman Drive, La Jolla, California 92093, USA. * These authors contributed equally to this work. Correspondence and requests for materials should be addressed to Z.L. (email: zhaowei@ucsd.edu).

Plasmonics, a field which has emerged within the last decade, has brought tremendous new opportunities to control light at scales previously thought impossible[1]. With the development of nanofabrication techniques, the dimension of plasmonic devices has been shrunk into the nanoscale[2], where the impact of quantum physics becomes important[3,4]. Various nanometre-scale plasmonic devices have shown new features that differ from traditional plasmonics[3–6]. Over the past few years, the impact of quantum effects on the linear properties of plasmonic devices has been examined from both experimental[4,7] and theoretical[8–10] points of view. The generation of nonlinearity through confinement of electrons has been extensively explored in the context of semiconductor quantum wells since the 1980s[11], where both the second-[12] and third-order nonlinear response[13] have been reported. Size-dependent nonlinear effects have been investigated in some of the previous studies on metallic nanoparticle systems[14,15]. However, the particle size was not small enough for the quantum size effect to play an important role in those systems and the significant contribution instead came from the hot electrons, which was a non-instantaneous nonlinear response.

Nonlinear optics has been an extensive research focus since the invention of the laser in the 1960s. The nonlinear interaction of light with matter itself leads to many interesting physical phenomena, such as harmonic generation[16], optical parametric oscillation[17] and optical soliton[18]. In addition, the nonlinear effects also lead to many useful practical applications. For example, frequency combs generated in optical cavities have been applied in metrology[19] and optical communication systems[20], and super-continuum generation through photonic crystal fibres is nowadays routinely used as a coherent broadband light source in imaging[21]. However, despite much effort and many years of advancement in this field, chip-scale-integrated nonlinear optics has yet to be realized owing to the lack of materials with exceptionally high nonlinear responses.

In this study, we investigate the optical Kerr nonlinear properties of a nanometre-scale gold quantum well, and find that the quantum size effect can lead to a giant nonlinear response, which shows a several-order enhancement compared to traditional nonlinear materials, making the thin metallic quantum well promising for on-chip-integrated nonlinear applications.

## Results

**Sample characterization.** Figure 1a shows a transmission electron microscopy (TEM) image of the cross-section of one of the 3 nm metal quantum well (MQW) samples (Methods). The confinement of the 'free' electrons of gold is realized by the large band gap of the two oxide materials ($Al_2O_3$ and $SiO_2$). As shown in the zoomed-in view of the TEM image (Fig. 1b), the gold film is continuous at the thickness of 3 nm. Although TEM figure can accurately tell us the thickness information of the MQW, atomic force microscopy (AFM) measurement is used to obtain the surface roughness right after the Au film deposition. Figure 1c shows the statistics of the film thickness variations through the AFM measurement, where we can see the surface roughness (root mean square) is about $\pm 0.25$ nm.

**Z-scan measurement.** To characterize the nonlinear optical properties of the MQW, z-scan experiment is performed (see schematic setup in Fig. 2a). Z-scan measurement is a sensitive and simple technique for measuring the complex nonlinear refractive index of a material[22]. Closed (normalized to open) and open aperture z-scan curve of the 3 nm MQW with 900 nm incident wavelength and 111 mW incident power is depicted using red circles in Fig. 2b,d, respectively. By fitting with the standard z-scan theory[22], the Kerr coefficient is calculated to be $(3.9-0.51i) \times 10^{-9}$ cm$^2$ W$^{-1}$ (Methods). To obtain the third-order susceptibility, the refractive index $n$ and

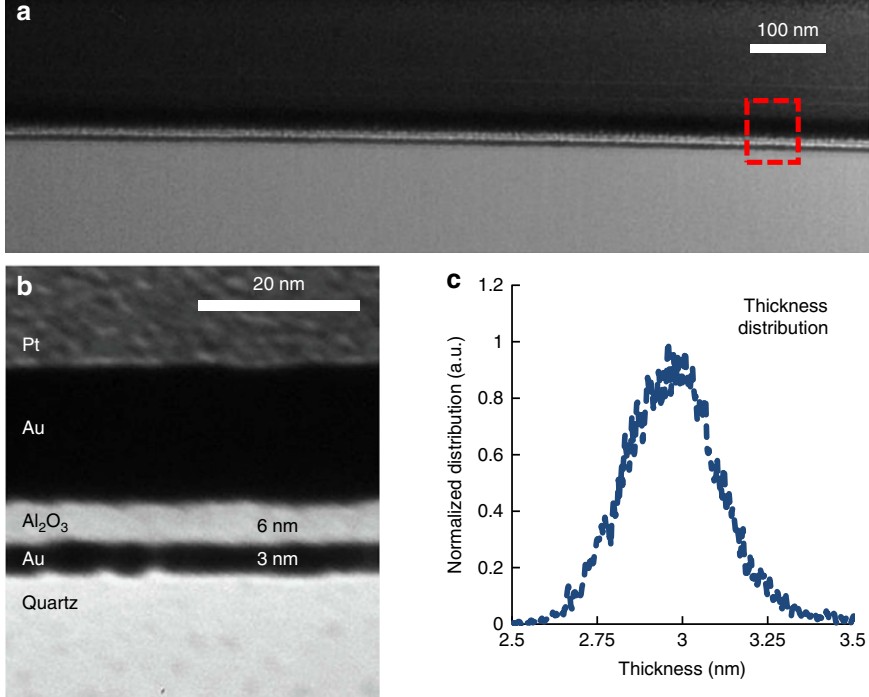

**Figure 1 | Sample characterization.** (**a**) Cross-sectional TEM image of the MQW sample prepared by focused ion beam (FIB). (**b**) A zoomed-in view of the MQW, that is, the bottom Au layer, indicates that the gold film is continuous. The thick Au and Pt layers atop were only used to facilitate the FIB process. (**c**) Statistics of the film thickness variations obtained from AFM measurement right after 3 nm gold film growth on the quartz substrate but before the $Al_2O_3$ film deposition. The surface roughness is around $\pm 0.25$ nm.

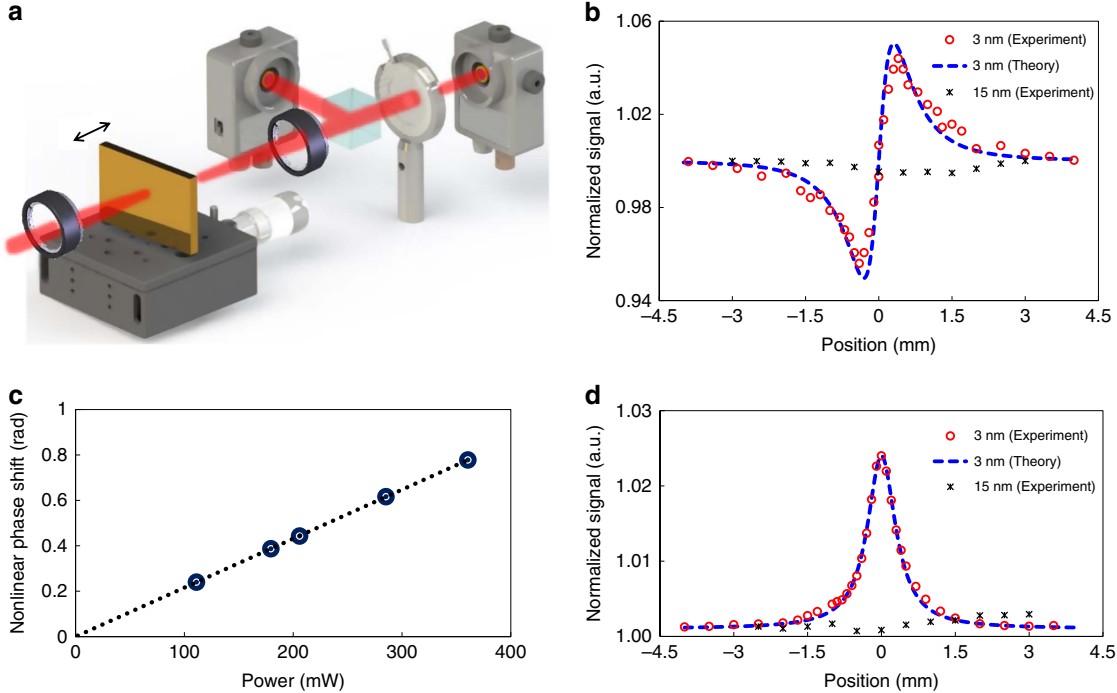

**Figure 2 | Z-scan nonlinear measurement.** (**a**) Schematic z-scan setup for third-order nonlinearity characterization. A femtosecond laser beam (Ti-sapphire laser, Mai Tai HP from Spectra-Physics, linearly polarized, 80 fs width and 80 MHz repetition rate) is focused onto the MQW sample through a lens ($f = 3$ cm). The raw laser beam diameter is about 3.5 mm, resulting in a Gaussian beam waist about 9 µm at the focus. The MQW is moved along the optical axis near the focus, two power metres are used to record the open and closed aperture z-scan signal through one collimation lens and a beam splitter. The closed aperture (**b**) and open aperture (**d**) z-scan curves for the 3 nm (red-circle) and 15 nm (black-cross) Au films using 900 nm incident wavelength with power of 111 and 690 mW. The closed aperture data are normalized to that of the open aperture. Z-scan signal of 3 nm sample is fitted by the standard z-scan theory (blue-dashed), from which the Kerr coefficient is extracted. (**c**) The measured nonlinear phase shift for different incident power. The dashed line is the mathematically linear fitting for the measurement data and nonlinear phase change is zero at zero intensity.

extinction coefficient $k$ is needed. For this purpose, reflection and transmission measurements are performed and a two-dimensional Newton's method together with a multilayer transmission algorithm is used to extract the $n$, $k$-values (Supplementary Note 1 and Supplementary Fig. 1). With $n = 0.4$ and $k = 3.7$, the third-order nonlinear susceptibility is calculated by[23]

$$\chi^{(3)} = \frac{\text{Re}\{n_2\} + i\,\text{Im}\{n_2\}}{283} n(n + ik) \qquad (1)$$

which gives $\chi^{(3)} = (0.49 + 2.0i) \times 10^{-15}$ m$^2$ V$^{-2}$. The magnitude of this value shows a four-order enhancement compared with the intrinsic value of bulk gold measured by four wave mixing[24]. The measured Kerr coefficient of the 3 nm MQW is six to eight orders larger than that of the traditional nonlinear mediums, such as fused silica[23]. It is worth noting that our sample is a flat metal film and no plasmonic mode (either localized or propagating) can be excited in the experiment. This rules out possible contributions from local field enhancement[25,26], but purely an intrinsic response of the MQW. It also indicates that the nonlinear response of the MQW may be further boosted by combining with additional plasmonic resonance structures with large field-enhancement factors. The incident power-dependent nonlinear phase changes is plotted out in Fig. 2c. Clearly, all data fall on a straight line crossing origin, confirming the detected nonlinear effect comes solely from the third-order Kerr response[23].

The physical reasons for such a giant nonlinear response are the quantum size effect and the high free electron density of the metal. According to the nonlinear theory[23], the third-order nonlinear susceptibility is proportional to $N(\mu_{mn})^4 / (\omega_{mn} - \omega - i\gamma_{mn})^3$, where $N$ is the density of free electrons, $\boldsymbol{\mu}_{mn} = -e\langle m|\hat{\mathbf{r}}|n\rangle$ is the dipole transition elements associated with the transition between state $n$ and $m$, the denominator is related to the resonant transition and $\gamma_{mn}$ is a damping term. The magnitude of $\chi^{(3)}$ is mainly determined by $N$ and $\boldsymbol{\mu}_{mn}$. Owing to the confinement effect, free electrons of the metal are quantized into subbands where their wave functions in the quantum well direction have an extension that is comparable to the well width $d$ (~several nanometres)[11]. The dipole transition elements associated to these inter-subband transitions are on the order of $e \cdot$ nanometre, which is much larger than those of the traditional nonlinear crystals (Supplementary Note 2). On the other hand, the $N$ of a typical MQW is on the order of $10^{28}$ m$^{-3}$, much higher than that of a typical semiconductor quantum well. These two facts imply that the nonlinear response of a MQW would be much larger than that of traditional nonlinear crystals and semiconductor quantum wells, making it among the most promising material systems for nonlinear applications.

As a direct comparison and verification, z-scan measurements for a thick gold film (15 nm gold film grown on quartz substrate with 6 nm Al$_2$O$_3$ on top) are also performed with 900 nm incident wavelength and 690 mW incident laser power. For a 15 nm gold film, the impact of quantum size effect is much smaller. As shown in Fig. 2b, no signal is detected at the closed aperture. For the open aperture, a change in normalized transmission $\Delta T = -0.133\%$ is recorded, which results in an imaginary part for the Kerr coefficient to be $3.4 \times 10^{-12}$ cm$^2$ W$^{-1}$. Using the literature data for the $n$ and $k$ for bulk gold[27] at 900 nm ($n = 0.17$ and $k = 5.72$), the third-order nonlinear susceptibility for 15 nm gold is calculated to be $\chi^{(3)} = (-9.1 + 0.35i) \times 10^{-19}$ m$^2$ V$^{-2}$, which agrees reasonably well with previous z-scan measurements of gold Kerr nonlinearity using femtosecond laser sources

($\leq 100$ fs)[28,29]. The measurement results are summarized and compared in Table 1. The $|\chi^{(3)}|$ of 15 nm gold film is slightly larger than that of the bulky gold measured by four wave mixing[24], which might relate to the weak confinement effect[30]. The slightly larger $|\chi^{(3)}|$ from ref. 28 using $z$-scan with 100 fs pulse compared with ref. 24 could come from using a wavelength closer to the interband transition. The much larger $|\chi^{(3)}|$ reported by using 5.8 ps pulse is due to the thermal nonlinear response.

**Wavelength dependence of the third-order nonlinearity.** The nonlinear response of the MQW is related to the transition of the quantized 'free' electrons between the subbands, which is determined by the MQW geometry. To explore this feature, $z$-scan experiment is performed on the 3 nm MQW by scanning the wavelength from 690 to 1,020 nm (limited by our femtosecond laser source). The measured nonlinear coefficients, both the real and imaginary parts, are plotted using red and green dots, respectively, in Fig. 3a. The measured nonlinear indices of the 3 nm MQW clearly show two resonance peaks in the measured spectral range, near 740 and 900 nm, respectively.

To fully understand the nonlinear optical properties of the MQW, a quantum electrostatic model[31] that is based on the self-consistent solution from Schrödinger and Poisson equations is used to calculate the eigen-state wave functions and eigen energies. As shown in Fig. 3b, there are eight quantized states supported by this MQW. More specifically, we found that the

transitions of the fifth to sixth and sixth to seventh eigen-energy states correspond to the experimentally observed resonant peaks around 900 and 740 nm, respectively (Supplementary Note 4 and Supplementary Fig. 2). After that, a standard approach that is based on perturbation theory[23] is adopted to calculate the $\chi^{(3)}$ of the MQW. Variations in the film thickness would lead to the broadening of the resonance feature of the material response. To take this effect into account, a Gaussian-broadening approach[32] is adopted by using the thickness variation data obtained from AFM measurement (detailed calculation and discussion can be found in Supplementary Notes 2 and 5). Finally, the calculated Kerr coefficients (both the real part and imaginary part) are plotted using dotted-line curves in Fig. 3a, which agree well with the experimental results.

**Nonlinear spectral broadening measurement.** Thermal effect can play a very important role in $z$-scan experiment[33]. The Kerr coefficient of gold film has been reported as high as $10^{-8}$ cm$^2$ W$^{-1}$ previously using $z$-scan method[34,35], where pulses with picosecond or even nanosecond duration were used. For such long pulse width, the dominant nonlinearity is the thermal effect[36,37]. In our experiment, pulses of only 80 fs are used and this effect can be neglected. To further exclude other contributions from thermal nonlinearity, the measurement of spectral broadening of a pulse due to self-phase modulation[38] is performed, as shown in Fig. 4. The reason is that optical Kerr

| | Measurement method | $\tau_{pulse}$ | $\lambda$ (nm) | $|\chi^{(3)}|$ (m$^2$ V$^{-2}$) | $\chi^{(3)}$ (m$^2$ V$^{-2}$) |
|---|---|---|---|---|---|
| 3 nm MQW* | Nonlinear spectral broadening† | 80 fs | 900 | $2.01 \times 10^{-15}$ | — |
| Bulk[24] | Four wave mixing | 200 fs | 800 | $2 \times 10^{-19}$ | — |
| 3 nm MQW* | $z$-scan | 80 fs | 900 | $2.06 \times 10^{-15}$ | $(0.49 + 2.0i) \times 10^{-15}$ |
| 15 nm gold* | $z$-scan | 80 fs | 900 | $9.1 \times 10^{-19}$ | $(-9.1 + 0.35i) \times 10^{-19}$ |
| 20 nm gold[28] | $z$-scan | 100 fs | 630 | $7.69 \times 10^{-19}$ | $(-7.68 + 0.43i) \times 10^{-19}$ |
| 20 nm gold[28] | $z$-scan | 5.8 ps | 630 | $7.58 \times 10^{-17}$ | $(-7.57 + 0.42i) \times 10^{-17}$ |

**Table 1 | Comparison of the $\chi^{(3)}$ values of gold from different measurements.**

*This work.
†Supplementary Note 3.

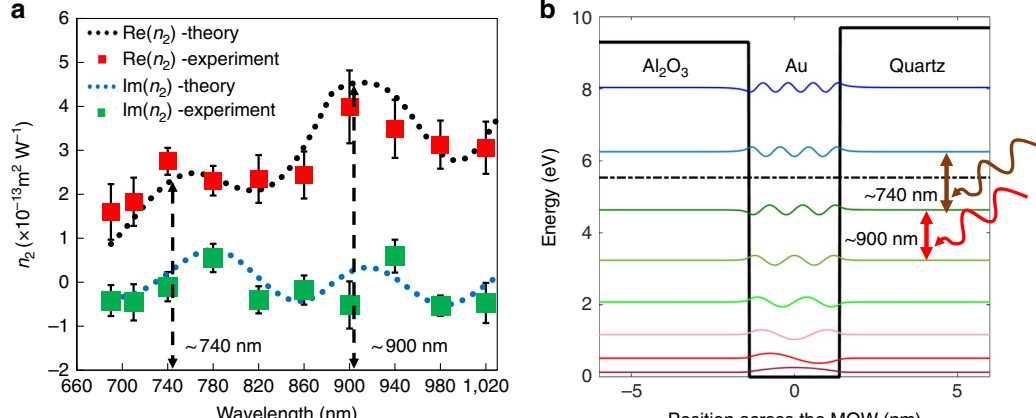

**Figure 3 | Wavelength dependence of the Kerr coefficient.** (**a**) Real (red dots) and imaginary (green dots) parts of the nonlinear indices measured for the 3 nm MQW as a function of incident laser wavelength. The fluctuations in multiple measurements at various locations are indicated by the error bars (s.d.). Two resonance peaks are found near 740 and 900 nm, respectively. Theoretical predictions (black and blue dotted lines) agree well with experimental results. (**b**) Calculated quantized energy states and their corresponding wave functions from the self-consistent calculation of the Schrödinger and Poisson's equations. For the 3 nm quantum well, the dipole dephasing time, which relates to the damping rate as $\tau = 1/\gamma$, is 7.5 fs and the effective mass of the electron is $0.35 \times m_e$, where $m_e$ is the free electron mass[31]. The Fermi energy level is plotted by dash-dotted black line. The transitions responsible for the two resonances observed in (**a**) are marked as the arrows in this figure as well.

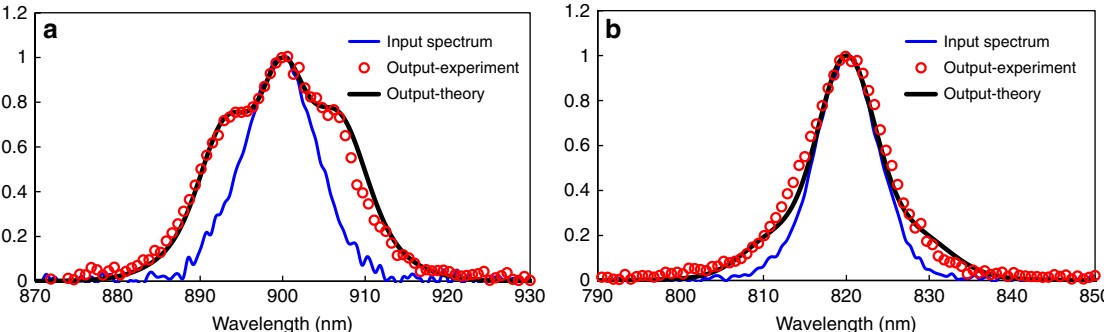

**Figure 4 | Nonlinear spectral broadening from the 3 nm sample.** A similar setup as the z-scan measurement is used with the sample placed at the focus for spectral broadening measurement, and the power meter at the open aperture branch is replaced by an optical spectrometer. Spectrum of an optical pulse (open red circles) measured after passing through the 3 nm MQW for incident wavelength at 900 (**a**) and 820 nm (**b**) using incident power of 650 and 750 mW. In the case of 820 nm, the self-phase modulation (SPM)-induced spectral broadening can only be seen at the spectral wings and the nonlinear phase shift is estimated to be $0.6\pi$. In the case of 900 nm, where the Kerr coefficient is at a resonance peak, the SPM-induced spectral broadening is much more obvious and the nonlinear phase shift is estimated to be $0.88\pi$. The input spectrum is shown by blue solid curves. Theoretical calculations of the transmitted spectrum based on the experimental value of $n_2$ measured from z-scan is plotted using black solid curves, which show excellent agreement with experimental data.

response is an instantaneous response, whereas athermal and thermal effects (Supplementary Notes 6 and 7) are much slower than Kerr response and would not lead to spectral broadening. The self-phase modulation-induced spectral broadening effect is proportional to the nonlinear phase $\mathrm{Re}(n_2) \times I \times k \times L_{eff}$, where $I$ is the light intensity, $k$ is the wave vector and $L_{eff}$ is the effective length[38]. Full simulation of optical pulse propagation[38] is performed by using the experimentally measure nonlinear coefficients and the corresponding transmitted spectrum are plotted using solid black lines in Fig. 4. As can be seen here, excellent agreement between theory and experiment is found, indicating that the third-order nonlinear effect measured from z-scan is indeed from the instantaneous Kerr response and the contribution from thermal effect can be neglected.

## Discussion

To summarize, we demonstrated giant Kerr nonlinear responses in MQW structures by the combination of the quantum size effect and the high free electron density of metal films. The measured nonlinear Kerr susceptibility $|\chi^{(3)}|$ reaches as high as $2.06 \times 10^{-15}\,\mathrm{m^2\,V^{-2}}$, showing a four order-of-magnitude enhancement compared with the intrinsic value of bulk gold[24,29]. Spectral broadening measurements confirm that the third-order nonlinear effect is an instantaneous Kerr response. Comparing with the localized surface plasmon resonance based nonlinear effects, the MQWs have the advantages of broadband operation and free of scattering (Supplementary Note 8). We expect that MQW structures can be easily combined with existing nano-photonic waveguides for integrated nonlinear optics[39], opening new opportunities for on-chip nonlinear optical applications with exceptionally high integration density.

## Methods

**Sample fabrication and characterizations.** The commercial magneto-sputtering machine AJA International in Nano3 facility at Calit2 was used for thin gold film growth and the qualities of the thin gold films were characterized by both TEM (FEI Tecnai) cross-section imaging and AFM (nanoScience Instruments) surface morphology mapping. Focused ion beam (FEI Scios DualBeam) milling was used to prepare the samples for later TEM imaging, as shown in Fig. 1 in the manuscript.

Reflection and transmission of the sample were measured by the commercialized spectrophotometer Lambda 1,050 system. Ti-sapphire laser (Mai Tai HP) was used for all the nonlinear measurement with Si Transimpedance Amplified Photodetectors (Thorlabs PDA100A) as the detector.

**Obtaining the nonlinear index from z-scan measurement.** This section shows how to obtain the nonlinear index of the 3 nm gold quantum well from measured z-scan curves in Fig. 2. From standard z-scan theory[22], the closed aperture transmittance (after normalization to the open signal) can be expressed as:

$$T_{close}(z) = 1 + \frac{4\Delta\Phi_0 x}{(x^2+9)(x^2+1)}, \tag{2}$$

where $\Delta\Phi_0 = k\,\mathrm{Re}(n_2)I_0 L_{eff}$, $k = 2\pi/\lambda$ is the wave vector and $x = z/z_0$, $I_0$ is the peak intensity, and $L_{eff} = (1 - e^{-\alpha L})/\alpha$ is the effective length with $\alpha$ as the linear absorption coefficient. $z_0 = k w_0^2/2$ is the diffraction length of a Gaussian laser beam with $w_0$ as the beam waist at focus. By best fitting equation (2) with the closed aperture z-scan curve in Fig. 2b, the maximum nonlinear phase shift $\Delta\Phi_0$ is determined to be 0.24 for the z-scan curves shown in Fig. 2 in the manuscript. By substituting all other parameters (laser intensity, effective length and wave vector), the value for the real part of the nonlinear index (Kerr coefficient) is finally calculated to be $\mathrm{Re}(n_2) = 3.97 \times 10^{-9}\,\mathrm{cm^2\,W^{-1}}$ at 900 nm for our 3 nm gold quantum well.

It is noteworthy that in the 3 nm quantum well sample, the thickness of the quartz substrate and the top $Al_2O_3$ layer are 300 μm and 6 nm, respectively. Considering that the nonlinear Kerr coefficient for quartz and $Al_2O_3$ are only $3.2 \times 10^{-20}$ and $2.9 \times 10^{-20}\,\mathrm{m^2\,W^{-1}}$, respectively, the contribution to the nonlinear phase shift from these two materials can be safely neglected. Indeed, the fact that the 15 nm gold sample has no detectable closed-aperture z-scan signal is also the experimental proof that the contribution from the substrate and top $Al_2O_3$ material is negligible compared with the 3 nm gold quantum well.

The normalized open aperture transmittance gives the information regarding the imaginary part of the Kerr coefficient. The normalized open z-scan curve can be expressed as:

$$T_{open}(z) = \sum_{m=0}^{\infty} \frac{(-q_0)^m}{(m+1)^{3/2}(1 + z^2/z_0^2)^m}, \tag{3}$$

where $q_0 = 2k\,\mathrm{Im}(n_2)I_0 L_{eff}$, with $m$ is an integer starting from 0. For the case of 3 nm, the transmission is increased by about 2.5% at the focus, corresponding to the phenomenon of saturable absorption. By fitting equation (2) to the z-scan curve, the imaginary part of Kerr coefficient for the 3 nm gold quantum well is calculated to be $-5.1 \times 10^{-10}\,\mathrm{cm^2\,W^{-1}}$. For 15 nm gold film, a very small decrease in the transmission ($\Delta T = -0.133\%$) is observed, corresponding to the two-photon absorption, which is typically observed for bulk gold. More specifically, by fitting the experimental data with z-scan theory, a value of $3.4 \times 10^{-12}\,\mathrm{cm^2\,W^{-1}}$ is obtained for the 15 nm gold film.

**Data availability.** The authors declare that the data supporting the findings of this study are available within the article and its Supplementary Information files.

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

## Acknowledgements

We thank Yunfeng Jiang for help on TEM sample characterization, and Qian Ma and Kangwei Wang for help on the spectrum broadening setup. We acknowledge financial support from the Office of Naval Research (ONR) Young Investigator Award (Grant Number N00014-13-1-0535) and the ONR MURI programme (N00014-13-1-0678).

## Author contributions

H.Q. and Y.X. designed and performed the experiments. H.Q. performed the sample growth and characterization. Y.X. performed the theoretical work. All authors analysed the data and wrote the manuscript. Z.L. supervised the research.

## Additional information

**Competing financial interests:** The authors declare no competing financial interests.

