## [Peer review file · Nature Communications]

Reviewers' comments:

Reviewer #2 (Remarks to the Author):

Dear Editor, dear Authors

I have carefully read the revised version of the manuscript from H. Qian et al. entitled "Giant Kerr Response of Ultrathin Gold Films from Quantum Size Effect" as well as the corresponding answer letter and Suppl. Information.

This paper reports interesting results regarding the enhancement of the optical Kerr susceptibility of a gold film when reducing its thickness, owing to quantum size effects. The results are sound and very fundamental. Experimental results are evidenced properly and supported by appropriate modelling. The work reported is, to my knowledge, original and valuable.

Most of my previous comments have been addressed by the authors, and I believe the paper is in a good way for publication, provided some points are modified and a few questions are answered.

1. Their rough estimation for comparing plasmonic enhancement of the susceptibility and the one they obtain with quantum confinement is sound. Let me nevertheless underline two points:
- for gold nanospheres in silica the modulus of the field enhancement factor is indeed lower than 10, but it may be much larger for other types of plasmonic nanostructures or coupled nanoparticles. The advantage of a thin gold film on this point is true when comparing with isolated plasmonic spheres.

- Scattering is fully negligible for small nanospheres, namely the size of which does not exceeds about 20 nm. What is presented here as an advantage of the thin film is then not really relevant. The evaluation of the nonlinear response for plasmonic nanospheres and for a thin film should be added in the Supplementary Information as a support of the authors' claim in the main text.

2. The authors give some good arguments (but some less good!) in their answer letter, but these arguments are not translated as changes in the main text. The latter still contains misleading explanations.

a. The argument based on the symmetry of the spectral broadening signal to support the insignificance of the electron distribution perturbation (athermal regime and further hot electron distribution) is robust and sufficient. But it raises several questions:

- Why in Fig. 4(a) the theoretical curve (black), which is supposed to be determined by considering only instantaneous mechanisms for the Kerr nonlinearity, is not perfectly symmetric?

- I really don't understand why the authors (beginning of Section 7 of the Suppl. Info.) describe the process of athermal electron distribution in the conduction band, and excitation of different quantized levels, as if they were completely independent processes. How can we distinguish these effects? If electron levels are quantized, then the conduction band itself is distorted. We cannot have two distinct description of the energy scheme of the electrons in the well! The authors should work on this part more rigorously.

- The authors claim in the Suppl. Info. that "the photon-induced athermal electron re-distribution has similar magnitude of impact on all those samples, i.e. excite electrons close to Fermi level, under similar power level." I don't agree: energy deposition is not the same (Beer-Lambert attenuation law) and the diffusion term in the electron dynamics becomes as significant as the film thickness is increased, thus fastening the local cooling of the excited electron gas.

- In the same paragraph, they write: "To further rule out other possibilities, we have performed

the spectral broadening experiment on 3 nm Au on Si substrate (Si substrate has much less free electron confinement compared with quartz substrate), and no spectrum broadening is found." It is very unclear. Why electron confinement of the substrate is involved here? And what should be the consequence of confinement? This absolutely needs a clearer explanation.

- In the answer letter, the authors put forward an argument regarding the relative population change of the quantized levels under pulse light absorption, by considering an "electron relaxation time for gold" of 7.5 fs. What does it correspond to? Is there a reference for this? It is known that electron-electron scattering time depends on electron energy relative to the Fermi level. As they base their analysis of non-instantaneity on this value, they should be more precise regarding the meaning of this "relaxation time" and refer to previous publication to support this choice of value.

b. The cumulative thermal effect is properly ruled out in the Suppl. Info. by good arguments.

c. On the contrary, the analysis of the single-pulse thermal effect is misleading because the authors use the term "hot electrons" for "thermal effect". The authors write in their main text: "For such long pulse width, the dominant nonlinearity is the "hot electron" effect, which is a delayed response by about 500 fs. In our experiment, pulses of only 80 fs are used, and the impact of "hot electron" effect can be neglected." The hot electrons release their thermal energy within 1 to 2 picoseconds by electron-phonon scattering. Then with 80-fs pulses I agree that the metal heating (in the classical phonon sense) has no influence on the signal their measure. However, the excited electrons could still have an influence! The authors can indeed rule out this influence because this process does not follow the temporal profile of the pulse and should then induce a non-symmetric spectral broadening profile. This last argument is very well described by the authors and is sufficient as a support. But I don't agree with the argument based on the comparison of the pulsewidth and the "delay time" for hot electrons, which is furthermore not well defined. What the authors should rather write is that the typical time needed to reach a hot electron gas at internal thermal equilibrium ranges from 0.1 to 1 picosecond and then, in their experiments with 80-fs pulses, does not contribute. Then, my advice for sake of clarity is that the authors should define much more precisely what they mean by "hot electrons" and by "thermal".

d. Before Table 1 it is stated that: "The much larger $|\chi(3)|$ reported by using 5.8 ps pulse is due to the hot electron contribution." I deeply believe that this is not true at all. The authors should explain this claim. The electron gas heating by such "long" pulses results in a quite low temperature increase and the pulsewidth being larger than the typical electron-phonon collision time the mean electron temperature during the pulse remains low. For me, the explanation is different: the high nonlinearity with 5.8-ps pulses comes from the so-called thermo-optical response of the metal (distortion of the band structure and displacement of the Fermi level with lattice expansion).

3. Grammar issues:

- In the main text, a "d" is missing at the end of "measure" in the sentence "Full simulation of optical pulse propagation is performed by using the experimentally measure nonlinear coefficients".

- In the Suppl. Info, the "s" of "leads" must be suppressed in the sentence "This excitation can leads to two possible changes".

Reviewer #3 (Remarks to the Author):

The manuscript reports on nonlinear properties of an ultrathin gold layer (3 nm) sandwiched between dielectrics. Such confinement brings into play new quantum properties, which result in huge numbers for the $\chi(3)$ susceptibility determined experimentally. Nice correlation between experimental and theoretical results is evident. Results are new, original and consistent. I think that the paper contains interesting material to be considered for publication in Nature Communications after revisions. I have some concerns on the material, which I formulated

through the comments below.

1. I think that the reasons to exclude the thermo-modulation effect from building up nonlinearity are well grounded. Even if the relaxation time of non-thermalized electrons to the thermalized state is not 500 fs, but 300 fs (Marini et al. , New J. Phys., v.15, 013033, 2013), still the pulse duration is too small to establish more or less competing third-order nonlinearity from the hot electrons distribution. However, presentation of data in Table I is very lopsided. For example, 100 fs pulses treatment of bulk gold properties in Baron et al., Phys Rev B, v. 91, 195412, 2015 provides values of $\text{Im}(\chi^3)$ in the order of $3 \cdot 10^{-16}$, what differs by less than one order of magnitude from MQW numbers. Data from Ref.24 as conditionally consistent with the last row in the table (22 nm gold, 3 ps pulses, SPP propagation) also provides the same order as Baron's paper but for the real part of χ^3 - $3 \cdot 10^{-16}$. So, I would still call the numbers presented in the manuscript as giant, huge, etc., but the 4th-orders of magnitude difference has to be mitigated in claims (abstract, conclusions).

2. Evaluation of the authors provide different signs for the $\text{Re} \chi^3$ in case of MQW and 15 nm layer. However, no any explanation or even constitution of this fact is made in the text. What I also find very interesting, that typically for gold layers (3 last rows in the table, and Ref.24 numbers) the real part of χ^3 is bigger, while for the MQW the imaginary part dominates with the same proportion. Is it due to some uncertainty in the refractive index restoration from the R-T measurements? It is worthy to discuss.

3. By the way, for me it looks like rather artificial that nonlinear properties are assigned to pure quantum effects, while the linear parameters restoration is 100% classical. Should it be revised synchronously? I have a concern towards the Fig.S1 results. R and T spectra are very monotonous in the whole range of characterization. However, refractive index curves have some fluctuations even starting from 700 nm, where $\text{Re}(n)$ is not very small to cause some instabilities in the restoration iterations. What I found suspicious is that the refractive index for 3 nm gold $0.4+i3.7$ has less imaginary part than the one for bulk gold ($0.17+i5.72$), however it is well known fact that the thinner the gold film - the bigger there are effective material losses (of bigger collision frequency if we are talking the Drude language). So my question is revolving again round the restoration procedure: can we trust these numbers? And what if the gold permittivity becomes a tensor in such thin layer (if any can be introduced at all)?

4. Some minor issues. i) From Fig. 1(b) and text it is not clear, which exactly samples were used in the z-scan characterization. I presume that such thick metal layers could have terminated any transmission through the samples. ii) Hardly I can call fused silica as nonlinear crystal, medium is physically more precise. iii) Does spectral broadening simulation (Fig.4) take into account small, but non-zero materials dispersion of gold film?

Reviewer #2

I have carefully read the revised version of the manuscript from H. Qian et al. entitled "Giant Kerr Response of Ultrathin Gold Films from Quantum Size Effect".

Following the two reviewers comments, the paper has been undoubtedly improved from several points of view. Then, I consider that *it could be published after the authors answer several points* which remain unclear or need further changes:

Author Response: We appreciate the reviewer's constructive comments and suggestions to improve the quality of our manuscript.

1. Ref. [5] of the 1st version is now set as Ref. [23] but is still quoted as Ref. [5] in the text.

Author Response: In the revised version, we cite the paper "Renger J, etc. Surface-Enhanced Nonlinear Four-Wave Mixing. Phys Rev Lett 2010, 104(4): 046803" [5] in order to have a direct comparison between the bulky χ^3 of gold measured from four-wave mixing and our χ^3 of 3 nm gold quantum well from spectral broadening measurement. We also cite the paper "Boyd RW, Shi ZM, De Leon I. The third-order nonlinear optical susceptibility of gold. Opt Commun 2014, 326: 74-79" [23] for more general comparison. To be more general, we now added Ref. [23] together with [5] in the manuscript.

2. In their answer to my point 1 of my first review, the authors don't really solve the problem. They insist on the fact that their system does not present any plasmonic enhancement of the field. But my point was that, because they present their results about very thin films as a very promising solution for integrated nonlinear optics, they should discuss the benefit of it versus the use of plasmonic gold which, for a lower metal quantity, can offer at least a similar nonlinear optical response in terms of magnitude. So, why thin gold films should be better than plasmonic gold nanoparticles for promising applications?

Author Response: We thank the reviewer to bring up this question. In the following, we provide a rough estimation regarding the nonlinear performance of quantum wells and plasmonic nanoparticles by using the same amount of gold.

Case (1): Plasmonic nanoparticles. We consider a dielectric medium of size $L \times L \times L$ with embedded gold nano spheres. Assume the volume fraction of gold nano sphere is p , the effective third order susceptibility can be obtained from the Maxwell-Garnett theory [1] as $\chi_{eff}^{(3)} = p|f|^2 f^2 \chi_m^{(3)}$, where $\chi_m^{(3)}$ is the intrinsic value for bulk gold and f is the field enhancement factor from localized surface plasmon resonance (LSPR). The Kerr phase modulation for such a medium is: $\chi_{eff}^{(3)} L = p|f|^2 f^2 \chi_m^{(3)} L$.

Case (2): Quantum wells. With the same transverse plane geometry ($L \times L$), it corresponds to multiple quantum well with total thickness of pL . The Kerr phase modulation can be written as: $\chi_{qw}^{(3)} pL = 10^4 p \chi_m^{(3)} L$, considering that the nonlinear susceptibility of gold quantum well is 4 orders larger than that of bulk gold.

In other nonlinear effects, such as degenerated four wave mixing or third harmonic generation, the nonlinear performance now is proportional to $|\chi_{eff}^{(3)} L|^2 = p^2 |f|^8 |\chi_m^{(3)}|^2 L^2$ for the composite medium [2], i.e. case (1) and $|\chi_{qw}^{(3)} pL|^2 = 10^8 p^2 |\chi_m^{(3)}|^2 L^2$ for the metallic quantum well, i.e. case (2).

One can see that the performance of the plasmonic particles heavily depends on the local field enhancement factor f . If $f=10$, the two cases have similar performance. According to the reference provided by the reviewer [1], f for gold is typically less than 10, which indicates that metal quantum well approach very likely possesses better nonlinear performance.

In addition, there are additional unique advantages for the metallic quantum wells:

(a) The high $\chi_m^{(3)}$ in quantum well system is intrinsically a broadband phenomena as demonstrated in our experiments. On the contrary, the field enhancement from LSPRs is usually a narrow band effect. Higher field enhancement is typically accompanied by narrower bandwidth.

(b) Metallic quantum well is a uniform medium, which is free of the scattering effect. Therefore, the metallic quantum well is a better option for specific integrated applications where scattering is not allowed.

(c) Metallic quantum well can also be combined with plasmonic nanostructure for better performance. For example, one can put plasmonic resonators close to quantum wells, so that the merits from both cases can be married.

We modify the last paragraph as follows:

...The measured nonlinear Kerr susceptibility $|\chi^{(3)}|$ reaches as high as $2.06 \times 10^{-15} \text{ m}^2/\text{V}^2$, showing a 4 order-of-magnitude enhancement compared to the intrinsic value of bulk gold^{5,6}. Spectral broadening measurements confirm that the 3rd order nonlinear effect is an instantaneous Kerr response. **Comparing with the localized surface plasmon resonance based nonlinear effects, the metal quantum wells have the advantages of broadband operation and free of scattering.** We expect that MOW structures can be easily combined with existing nano-photonics waveguides for integrated nonlinear optics³², opening new opportunities for on-chip nonlinear optical applications with exceptionally high integration density.

Reference:

[1] chapter 15 of the book "Nonlinear optical properties of matter: From molecules to condensed phases", Series: Challenges and Advances in Computational Chemistry and Physics, Vol. 1, edited by M. G. Papadopoulos, J. Leszczynski and A. J. Sadlej (Springer, New York, 2006)

[2] Ricard, D., Ph Roussignol, and Chr Flytzanis. "Surface-mediated enhancement of optical phase conjugation in metal colloids." *Optics letters* 10.10 (1985): 511-513.

3. In their answer, the authors write that hot electrons and accumulated thermal effects are non-instantaneous effects and then can contribute to the z-scan signal and not to the spectral broadening signal. I don't understand: in both cases, there is only one pulse which both creates the nonlinear response and feels the induced modifications of the material optical properties. The fact that non-instantaneous phenomena can or not be felt only depends on the duration of the pulses which are used, whatever the type of measurement.

Author Response: We thank the reviewer for this comment. The difference in z-scan and spectral broadening measurement is that the former measures the spatial intensity dependence of the refractive index, while the later measures the temporal intensity dependence of refractive index [3]. In the case of only one pulse, we agree with the reviewer that in both cases the non-instantaneous phenomena can or not be felt only depends on the duration of the pulses which are used. However, periodic pulses are used in both z-scan and spectral broadening experiments. One needs to consider the impact of repetitive effects, or the accumulated thermal effects as what we termed in the manuscript.

In the case of z-scan, the accumulated thermal effect may lead to a spatially-dependent refractive index change because of the invariant laser intensity profile. Therefore, this type of accumulated thermal effects can express similar instantaneous Kerr response (which only comes from the single pulse itself). While a stabilized accumulated thermal effect due to the repetitive pulse train is constant in the temporal domain and is independent on the each pulse shape. In this case, the accumulated thermal effect cannot contribute to the spectral broadening. This is the motivation of the spectral broadening measurement. We also added the above explanation in the supplementary information in section 9.

In addition, the answer in the following question (comment 4) also proves the values of the spectrum broadening measurement.

Reference:

[3] Agrawal, Govind P. Nonlinear fiber optics. Academic press, 2007.

4. There is still a controversy about the involvements of "hot electrons". The authors refer to former Ref. [5] as well as the article in Phys. Rev. B 1994 of Sun et al. to state that the contribution of hot electrons only appears after 500 fs. This is all but true, because:

- 500 fs is not the turn-on time of the hot electron contribution, but it is the characteristic time for the establishment of a thermalized distribution of hot electrons after the initial athermal regime.
- This characteristic time has been shown later by F. Vallee to decrease with decreasing laser pulse duration at fixed pulse energy, and also to decrease with increasing pulse energy at fixed pulsewidth.
- The athermal regime for the electron distribution, even if we cannot rigorously use the word "hot electrons" due to the fact that a temperature cannot be defined in this out-of-equilibrium state, has a strong influence on the metal optical properties and contributes to the value of the 3d-order nonlinear susceptibility, as was clearly shown by Guillet et al. (Phys. Rev. B 79, 045410 (2009)). It is easily understandable that if you modify strongly the electron distribution due to absorption, especially around the Fermi level, then the dielectric function is strongly modulated, especially close to the interband transition threshold. So, contrarily to what the authors write in their answer, for pulses shorter than 100 fs the effects of the photo-induced athermal electron distribution (maybe better term than "hot electrons") is definitely felt by the pulse. What the authors call "a delayed response" regarding hot electrons is the response of the electron distribution when a thermal behavior is recovered, which indeed can take several hundreds of femtoseconds. But before this, the distribution is strongly affected by the pulse, and the effect is instantaneous, so felt by the pulse itself.

The consequence is that the authors should justify further the fact that the contribution of this out-of-equilibrium electron distribution can be neglected. The argument based on time is not proper. In their answer (2) to my previous point 9, they provide an interesting explanation about the number of electrons which are affected by the photon absorption. However, this is not really relevant as the perturbation is not homogeneously distributed among the whole electron gas whatever electron energy. They should justify further, maybe by considering that the wavelengths at which they carry out the measurements are far from the interband transition threshold. But remember that in the ultrafast athermal regime the perturbation can impact the interband transitions even far below their threshold usually defined in the stationary regime, due to the non-thermal distortion of the electron distribution down to electron energies of $E_F - E_{\text{photon}}$ (disregarding multiphoton processes). So, to rule out any possibility of athermal electron distribution implication, more rigorous arguments need to be provided.

Author Response: We thank the reviewer for this insightful comment. We agree with the reviewer that photon-induced athermal electron distribution is felt by the pulse, and it is necessary for further explanation to justify the fact that the contribution of this out-of-equilibrium electron distribution can be neglected.

Direct support for our argument comes from the fact that $\chi_m^{(3)}$ of the 15 nm gold film (our measurement) and the bulky gold (from literature) is 3 to 4 order-of-magnitude smaller than the 3 nm gold quantum well. We anticipate that the photon-induced athermal electron re-distribution has similar magnitude of impact on all those samples under similar power level. Therefore, the high $\chi_m^{(3)}$ in 3 nm gold quantum well should dominantly come from quantum effect, rather than the athermal electron distribution. To further rule out other possibilities, we have performed the spectral broadening experiment on 3 nm Au on Si substrate (Si substrate has much less free electron confinement compared with quartz substrate), and no spectrum broadening is found.

Another support can be described from the spectral broadening measurement. The excitation of an optical pulse would drive the electron distribution into out-of-equilibrium states. This excitation may lead to two consequences:

- (1) the transition of electrons between two quantized energy states, which is unique to the metal quantum well, leading to a population change of each quantized states;
- (2) the perturbation of electron energy distribution within the same conduction band, as is discussed in the reference [4] provided by the reviewer (we also cite this paper in the manuscript).

Rather than developing a complete and comprehensive model to describe the dynamics of athermal electron distribution (definitely worth for a separate publication), its impact actually can be estimated from the spectral broadening measurement. The reason is as following:

Optical Kerr nonlinearity is an instantaneous response. An optical pulse sees a nonlinear phase shift profile exactly following the shape of the pulse itself inside a Kerr medium. For an input of symmetric Gaussian pulse, the change in the refractive index felt by the pulse itself is also symmetric Gaussian shape. As a consequence, the laser pulse would develop a spectral broadening that is symmetric.

The photon-induced athermal electron redistribution is not an instantaneous process as Kerr effect. The excitation process of an electron by absorbing an incident photon can be viewed as an instantaneous process. However, the relaxation process is non-instantaneous. Therefore, an optical pulse would not experience a symmetric refractive index change in temporal domain. As a consequence, the pulse spectrum would not develop a symmetric broadening in this case.

To illustrate the above arguments, we first consider case (1), which we think is more important to our metal quantum well. We assume an 80 fs laser pulse excitation, and change in the electron population density ΔN (electrons excited from a lower quantized energy state near the Fermi level to a higher one by absorbing an incident photon) can be characterized by the following equation:

$$\frac{d\Delta N}{dt} = \frac{A}{h\nu} - \frac{\Delta N}{\tau},$$

where ν is the frequency of incident photon, A is the absorbed energy density, and τ is the electron relaxation time for gold (~ 7.5 fs). This equation is solved and the corresponding ΔN is plotted using red-curve in Fig 1(a). The incident pulse profile is shown by the blue-curve in the same figure as a

comparison. As shown here, the change in electron distribution is very similar to the incident pulse shape, but it is delayed by the relaxation time.

In the case of Kerr interaction, the refractive index change is proportional to the incident pulse intensity [5] (blue-curve in Fig. 1(a)). In the case of athermal excitation, the refractive index change of the metal quantum well can be related to the population probability change of state ρ_{mm} (red-curve in Fig. 1(a)) through the following equation [5], considering that the linear material property in our measurement region is mainly contributed from the quantized free electron energy state.

$$\chi^{(1)}(\omega) = \frac{N_0}{\epsilon_0 \hbar} \sum_{nm} \rho_{nm} \left[\frac{\mu_{nm} \mu_{nm}}{(\omega_{nm} - \omega) - i\gamma_{nm}} + \frac{\mu_{nm} \mu_{nm}}{(\omega_{nm} + \omega) + i\gamma_{nm}} \right]$$

Figures 1(b-d) show the input and calculated spectrum of the output pulse for different percentages of athermal contribution, where in all cases the maximum phase shift due to refractive index change is taken to be 0.88π . The input spectrum is shown using dashed-green curve. The output spectrum considering athermal contribution is shown by the red curve, while the output spectrum due to pure Kerr effect is shown by the blue curve. As shown in (b), even 10% of athermal contribution leads to considerable asymmetry in the pulse spectrum. Since the experimentally measured pulse spectrum is almost symmetric-broadened (Fig.4 in the manuscript), athermal contribution in our case should be well less than 10%.

Figure 1 | Spectral broadening simulation based on the pure intrinsic Kerr response and athermal contribution. (a) Input power profile (dash-blue) and electron population change with time (red). (b-d) Spectral broadening simulation results considering different contribution percentage of the athermal process, where less than 10% contribution from athermal can be estimated considering the symmetric broadening from experimental measurement.

For case (2), it has been discussed in Ref. [4] that the out-of-equilibrium energy distribution of electrons takes about few hundreds of femtoseconds to form a thermalized electron distribution by electron-electron, electron-phonon interaction. In this case, the excitation and relaxation process would be much more asymmetric as compared to case (1), which would lead to a much more asymmetric pulse spectrum change. Therefore, the contribution of the athermal process can be safely ruled out by our spectral broadening measurement.

Following the suggestion of reviewer, we also have added some of the above discussion in the supplementary information in section 7.

Reference:

- [4] Yannick Guillet, Majid Rashidi-Huyeh, and Bruno Palpant. “Influence of laser pulse characteristics on the hot electron contribution to the third-order nonlinear optical response of gold nanoparticles”. *Phys. Rev. B* 79, 045410 (2009).
- [5] Boyd RW. Nonlinear Optics, 3rd Edition. *Nonlinear Optics, 3rd Edition* 2008: 1-613.

Reviewers' comments:

Reviewer #2 (Remarks to the Author):

Dear Editor, dear Authors

I have read the last answer letter as well as the revised manuscript and Suppl. Info from the authors.

They have properly accounted for my remarks and they have subsequently modified their manuscript, which for me is much clearer now. They have kept the relevant evidences for their interpretation and precised why they can rule out any effect of the athermal electron distribution as well as of further metal heating.

As I'm now convinced by the paper, I agree for it being published.

I'm also satisfied to have contributed to its improvement along these last months.

Reviewer #3 (Remarks to the Author):

In overall I'm satisfied with the discussion, and verbose answers provide a lot of clarifications. Nevertheless, I still have a point, which I cannot understand (or accept) from what is reported in the article. Namely, pure quantum interpretation of the effect. If one look in Fig.3, then two peaks are well associated with the "quantum transitions" 5-6 and 6-7. However, away from these resonances the average (or background) numbers for n_2 are still extremely (remarkably) high - somewhere at the 10^{-13} level. It means that even far from any "quantum transitions"-caused χ_3 enhancement the system exhibits a very high nonlinear response. Can this effect be explained quasi-classically just by accounting for inverse-thickness dependent nonlinearity in the limit of 3-4 nm gold layer? Or may be the rationale is to combine broad flat inverse-thickness dependence of χ_3 (n_2) with local QW-levels related resonances in interpretation of the giant Kerr response?

Reviewer #3

In overall I'm satisfied with the discussion, and verbose answers provide a lot of clarifications. Nevertheless, I still have a point, which I cannot understand (or accept) from what is reported in the article. Namely, pure quantum interpretation of the effect. If one look in Fig.3, then two peaks are well associated with the "quantum transitions" 5-6 and 6-7. However, away from these resonances the average (or background) numbers for n_2 are still extremely (remarkably) high - somewhere at the 10^{13} level. It means that even far from any "quantum transitions"-caused $\chi^{(3)}$ enhancement the system exhibits a very high nonlinear response. Can this effect be explained quasi-classically just by accounting for inverse-thickness dependent nonlinearity in the limit of 3-4 nm gold layer? Or may be the rationale is to combine broad flat inverse-thickness dependence of $\chi^{(3)}$ (n_2) with local QW-levels related resonances in interpretation of the giant Kerr response?

Author Response: We thank the reviewer for bringing up this question.

First of all, we are not quite sure what the reviewer means by saying "inverse-thickness dependent".

- (1) Assume that the reviewer means $\chi^{(3)}$ is proportional to $1/d$ or $1/d^2$, so that the thinner the thickness is, the bigger the nonlinear response is. In this way, we cannot agree with the reviewer on this, and the quantum nonlinear theory that we used¹ certainly does not predict this result.
- (2) Assume that this "inverse-thickness" means a certain thickness induced quantum size effect determines the nonlinear response (including its wavelength dependence and resonance feature), and then a fluctuation in the thickness would lead to the broadening of such resonances, which would contribute to the experimental result of a relatively large "background" numbers for n_2 or $\chi^{(3)}$. Then we would agree with the reviewer on this point. In supplementary section 5, we have considered the thickness variation in the quantum nonlinear calculation, and the relatively large n_2 or $\chi^{(3)}$ away from the transition is coming from the broadening of the quantum states resonance transition.

Note that the thickness variation is not the only reason for a broad resonance for the nonlinear response. The relative small dipole dephasing time of gold here also contributes to this effect¹. In typical semiconductor quantum wells, this number is much larger than metal, and consequently, the resonance feature for semiconductors are much more pronounced with a much sharper resonance. In this case, the nonlinear response would be much weaker away from resonance. Also note that the broadening of the transition resonance will decrease the resonant peak value.

Therefore, to answer reviewer's question, this broadband nonlinear response (or the relatively large average number of n_2 far away from resonance) of our metallic quantum well is due to two facts: (1) thickness fluctuation and (2) small dipole dephasing time. To clarify this point, we have added following sentences at the end of supplementary information section 5.

"..... The fluctuation in the thickness would lead to the broadening of the transition resonances, which would contribute to the experimental result of a relatively large numbers for n_2 or $\chi^{(3)}$ away from the resonance. Note that the thickness variation is not the only reason for a broad resonance for the nonlinear response. The relative small dipole dephasing time of gold here also contributes to this effect".

1. Boyd RW. *Nonlinear Optics, 3rd Edition*. Academic Press, 2008.

REVIEWERS' COMMENTS:

Reviewer #3 (Remarks to the Author):

I read and accepted the authors position explaining the broadening of chi-3 enhancement effect. In spite that I am not fully convinced in the quantum origin of the observed effect I think that the article can stimulate fruitful discussion on this complicated and ambiguous point. Hopefully I can contribute to this discussion too. So I recommend the manuscript for publication in Nature Communications.